# Inheritance of Early and Late Ascochyta Blight Resistance in Wide Crosses of Chickpea

**DOI:** 10.3390/genes14020316

**Published:** 2023-01-26

**Authors:** Abdulkarim Lakmes, Abdullah Jhar, Adrian C. Brennan, Abdullah Kahriman

**Affiliations:** 1Department of Field Crops, Harran University, Sanliurfa 63100, Turkey; 2Biosciences Department, Durham University, Durham DH1 3LE, UK

**Keywords:** Ascochyta blight, *Cicer arietinum*, wild crop relatives, pathogen resistance, quantitative trait locus, polygenic inheritance

## Abstract

Chickpea (*Cicer arietinum*) is a globally important food legume but its yield is negatively impacted by the fungal pathogen Ascochyta blight (*Ascochyta rabiei*) causing necrotic lesions leading to plant death. Past studies have found that Ascochyta resistance is polygenic. It is important to find new resistance genes from the wider genepool of chickpeas. This study reports the inheritance of Ascochyta blight resistance of two wide crosses between the cultivar Gokce and wild chickpea accessions of *C. reticulatum* and *C. echinospermum* under field conditions in Southern Turkey. Following inoculation, infection damage was scored weekly for six weeks. The families were genotyped for 60 SNPs mapped to the reference genome for quantitative locus (QTL) mapping of resistance. Family lines showed broad resistance score distributions. A late responding QTL on chromosome 7 was identified in the *C. reticulatum* family and three early responding QTLs on chromosomes 2, 3, and 6 in the *C. echinospermum* family. Wild alleles mostly showed reduced disease severity, while heterozygous genotypes were most diseased. Interrogation of 200k bp genomic regions of the reference CDC Frontier genome surrounding QTLs identified nine gene candidates involved in disease resistance and cell wall remodeling. This study identifies new candidate chickpea Ascochyta blight resistance QTLs of breeding potential.

## 1. Introduction

Chickpea (*Cicer arietinum* L.), belongs to the Fabaceae (Leguminosae) family and Papilionoideae subfamily. It is self-pollinated, diploid (2*n* = 2*x* = 16 chromosome), with a 732 Mbp genome size [1,2]. Turkey is considered one of the main chickpea-producing countries in the world, where it is also the most produced food legume, but other important production regions extend through India, Pakistan, Canada, Mexico, and Australia (https://www.fao.org/faostat/en/, accessed on 6 March 2022). Chickpea is a nutritionally balanced food and is therefore important for subsistence and food security in the developing world [3,4].

Crop production is challenged by a range of biotic factors, a major pathogen of chickpeas being Ascochyta blight caused by *Ascochyta rabiei* (Pass.) Labr. Ascochyta blight, which can cause up to 100% yield losses in the field when high humidity conditions are present [5,6,7]. Agricultural practices such as the removal of crop debris between sowings, crop rotations, and the application of fungicides may help to prevent spreading of the diseases, but developing resistant varieties would be a promising method to overcome the problem. There has been some success in strategies to improve genetic control of Ascochyta blight, especially through the introduction of resistance alleles from wild crop relatives of chickpea [4,6,8]. There is considerable natural genetic variation across world gene bank collections and in the wild, but a more systematic exploration of wild chickpea collections for their plant breeding utility is needed [4,9,10,11].

In this study, we examined the inheritance of resistance to Ascochyta blight in two wide-cross chickpea families (Gokçe × Oyali-084, Gokçe × Karab-092) under field conditions in Southern Turkey. The families represent interspecific crosses between cultivated *C. arietinum* and wild-sampled accessions of *C. reticulatum* and *C. echinospermum*, respectively. Previously, these families have been used for quantitative genetic analysis and quantitative trait locus (QTL) mapping of traits related to flowering and seed yield [12,13]. The DNA markers found to be associated with resistance in this study could be used to accelerate chickpea breeding with marker-assisted selection. The study objectives were (i) to develop efficient and accurate screening methods for quantitative genetic analysis of Ascochyta blight resistance under typical growing conditions in southern Turkey, (ii) to combine resistance data with single nucleotide polymorphism (SNP) genotype data to genetically map QTLs associated with disease resistance, and (iii) to interrogate the genomic regions near to QTLs for the presence of candidate resistance gene alleles that could be responsible for improved resistance.

## 2. Materials and Methods

### 2.1. Field Experiments

Two chickpea interspecific cross families consisting of 305 lines in total were used in the experiment (160 lines of Gokçe × Oyali-084 and 145 lines of Gokçe × Karab-092). Hereafter, these families are referred to as GO and GK, respectively. To develop the families, crosses were made between Gokce, the cultivated species *C. arietinum*, and wild genotypes belonging to *C. reticulatum* and *C. echinospermum* species recently collected from their native ranges in Turkey [9]. The wild and cultivated varieties show many differences that include the wild species being small-seeded, purple-flowered, prostrate with indeterminate growth habits, spiny stems, and greater disease resistance. Each F_1_ line was grown in the glasshouse to obtain approximately 200 F_2_ seeds. Then, F_2_s were advanced by selfing, by bulking five seeds per line per generation. This method of advancement was chosen over single-seed descent to maintain as many lines as possible. The seeds from the F_2:5_ generations were used for the field experiments of this study.

The 305 F_2:5_ lines were grown under field conditions at the Field Experiment Station of Harran University, Turkey (37.10 N 39.06 E, 550 m altitude, “hot dry summer” CSA Köppen climate type). Five seeds from each line were randomly chosen, nicked by a nail-clipper to promote germination, and planted in the field on 5 February 2020. The families were sown in an augmented design consisting of four unreplicated blocks per family because of the limited amount of seeds of experimental lines. Each block consisted of up to 40 lines in addition to four checks. These checks were the resistant wild parent, the susceptible cultivated parent, the resistant check line ILC9279, and the susceptible check line ILC1929. Natural infection from soil was expected but we also ensured infection exposure at the first true leaf stage by inoculating with infected crop debris of the previous season that was scattered by hand within the plot. Furthermore, we increased the probability of infection with increased humidity using sprinkler irrigation on the day before and after inoculation.

Disease assessment of families, parents, susceptible, and resistant checks was then conducted weekly from one week after inoculation, recording the severity of symptoms, based on the commonly used 1 to 9 scale for chickpea resistance [5]. The phenotype scores were as follows: 1 no visible lesions; 2 lesions visible by close examination; 3 a few lesions visible that were easily seen; 4 many lesions visible, but lesions have not caused irreparable damage to the plants; 5 large lesions on stem or leaves, some leaf and stem girdling, but the plant is still alive; 6 many large lesions on stem and leaves, moderate stem and leaf girdling, but the plant will probably survive; 7 many large lesions on stem and leaves, stem and leaf girdling common, the plant may or may not die but will produce few seeds; 8 large lesions on stem and leaf common, stem and leaf girdling common, the plant probably will die; and 9 infection severity such that the plant is dead or dying. Six weekly measures were completed until the most susceptible genotypes had died. 

### 2.2. Trait Analysis

Statistical analysis was performed with Excel 2019 (Microsoft, Redmond, DC, USA), SPSS v23 (IBM Armonk, NY, USA), and R software (https://www.R-project.org/, accessed on 10 August 2021). The six disease scores for each line and check were used to calculate the overall disease severity for each successive disease assessment using the formula: Disease severity = (Sum of all disease ratings × 100)/(Total no. of ratings × maximum disease grade).

Area Under Disease Progress Curve (AUDPC) was calculated for each line and check for each successive disease assessment using the “Agricolae” R package (https://CRAN.R-project.org/package=agricolae, accessed on 6 July 2022). The values for each weekly measure, AUDPC, and severity were adjusted for block effects using augmented design with the “plantbreeding” R package (http://plantbreeding.r-forge.r-project.org, accessed on 6 July 2022).

Summary statistics were calculated for all adjusted measures and the distribution of measures for each family was tested for normality using Shapiro–Wilk tests. Spearman rank correlation was used to compare adjusted severity and adjusted AUDPC measures. AUDPC differences between line types; first parent, second parent, resistant check, susceptible check, family, and between families were tested with Mann-Whitney-Wilcoxon and Kruskal–Wallis tests followed by Dunn’s post-hoc comparisons with Benjamini-Hochberg multiple testing correction.

### 2.3. Genotyping

The genotyping of these two mapping families is described in a related study of seed weight [12]. Briefly, genomic DNA was extracted from dried leaf samples collected in the field and genotyped for a total of 60 Kompetitive Allele Specific PCR (KASP) markers targeting single nucleotide polymorphisms mapped to the NCBI chickpea reference genome CDC Frontier (BioSample: SAMN02981489). The DNA extraction and genotyping services were provided by Biosearch Technologies (Hoddesdon, UK). A total of 160 GO lines and 152 GK lines were genotyped, of which 150 GO lines and 142 GK lines had also been phenotyped for Ascochyta resistance.

### 2.4. Quantitative Trait Locus Analysis

Physical map locations in Mbp units against the reference CDC Frontier chickpea genome from NCBI (BioSample: SAMN02981489) were used for QTL analysis. The traits that were analyzed were disease scores at each time period, AUDPC, and severity. The QTL mapping using the R package qtl2 (https://cran.r-project.org/web/packages/qtl2/index.html, accessed on 12 May 2020) following published methods [12]. The analysis was performed with a calculated matrix of genotype probabilities at 1 Mbp intervals including the option of kinship correction leaving out the focal chromosome (“loco”). Data permutations (5000) were used to calculate the 95 % logarithm of odds (LOD) threshold to identify significant QTL LOD peaks. The effect sizes of significant QTL peaks were measured by qtl2 and PVE was measured by transforming LOD scores according to the formula 1 − 10^(−2 LOD/*n*), where *n* is the number of measured phenotypes. The QTL LOD maps and genotype × phenotype plots of significant QTL peaks were plotted with qtl2.

### 2.5. Candidate Gene Search

A 200k bp region centered around each significant QTL peak LOD position was chosen due to the relatively high linkage disequilibrium that has been reported for the chickpea genome [14]. These QTL regions were searched for annotated genes on the chickpea CDC Frontier reference genome on the PulseDB resource (https://www.pulsedb.org, accessed on 9 September 2022). The PulseDB sequence tool was used to view the gene annotations present in each 200k bp region, and then information about UniProt annotation, and SwissProt or TrEMBL homologs in other plant species were extracted for each gene. The UniProt pages (https://www.uniprot.org/uniprotkb/ accessed on 9 September 2022) for top homologs in the model plant *Arabidopsis thaliana* were consulted and genes with putative functions in defense or cell wall remodeling were noted.

## 3. Results

### 3.1. Ascochyta Resistance Expression

Disease scores, severity, and AUDPC data are available in Appendix A. Summary mean and standard deviation values for disease scores at each stage, severity, and AUDPC are presented in Table 1. Disease severity and AUDPC values were highly correlated according to Spearman’s ranked test (rho > 0.99) with equivalent results so only the results for AUDPC are presented. Measures were non-normally distributed in each family, as shown for AUDPC in Figure 1. The family AUDPC distributions were broad, with a long tail of individual lines with high resistance as well as some individuals that were more susceptible than the cultivated parent. These results indicate both polygenic and transgressive inheritance of Ascochyta resistance.

Changes in disease progression across each measurement period for each family, their parents, and their respective check lines are presented in Figure 2. Disease severity increased over time but the relative rankings of each line type remained constant. Comparisons of AUDPC between the experiments involving each family and their respective lines showed that only one family differed significantly between experiments (Appendix A). Susceptible and resistant checks showed the highest and lowest AUDPC as expected. The Gokce parent was the next most susceptible line and the wild parents were the next most resistant lines confirming expected differences between the parental lines. Differences between parental lines and their most similar resistant or susceptible check were not significantly different according to non-parametric tests (Appendix A). Each family was significantly more susceptible compared to its wild parent line but insignificantly different from the cultivated Gokce parent. All other within-family paired-line comparisons were significant.

### 3.2. Ascochyta Resistance Quantitative Trait Loci

Previously published genotype data are available in the Appendix A of Lakmes et al. (in press). A total of four significant Ascochyta resistance QTLs were identified across the two families. For the GO family, one QTL was found for the last 42 days disease score at 3.22 Mbp position on chromosome 7 with a PVE of 9.04 % (Table 2, Figure 3a). This genetic map region also showed the greatest LOD scores for 14 days disease score, AUDPC, and severity, although these peaks were slightly below the 95 % confidence threshold to call QTLs (Appendix A). The additive effect size of the 42 days disease score was -0.03 in the direction of the wild O allele, while the dominant effect size was 0.79 (Table 2, Figure 4). For the GK family, a distinct significant QTL was found for each of the disease scores at 7 days, 14 days, and 21 days. The 7 days disease score QTL was at 30.81 Mbp positions on chromosome 3; the 14 days disease score QTL was at 4.04 Mbp position on chromosome 6; and the 21 days disease score QTL was at 33.36 Mbp position on chromosome 2 (Table 2, Figure 3b). The PVE of these QTLs ranged from 9.39 to 9.49 %; the additive effects ranged from -0.33 to 0.22 in the direction of the O allele; and the dominant effects ranged from 0.44 to 0.64 (Table 2, Figure 4). Relatively few homozygote genotypes were observed at the 14 days QTL (Figure 4). High but insignificant QTL peaks were also observed at these locations for disease scores at other measurement times and for AUDPC and severity (Figure 3b and Appendix A).

### 3.3. Ascochyta Resistance Candidate Genes

The lists of all genes found within QTL regions, including their locations on the CDC Frontier genome assembly, their closest *A. thaliana* or other plant species homolog, and their primary InterProt annotation are in Appendix A. Some of the genes had uncharacterized functions meaning that their role in Ascochyta resistance cannot be ruled out. Genes with putative functions relevant to disease resistance have been summarized in Table 3. The GO 42 days disease score QTL region contained 25 annotated genes, of which two genes, *Ca_03156* and *Ca_03143*, had potential roles in disease resistance and one gene, *Ca_03139*, had a role in cell wall remodeling. The GK 7 days disease score QTL on chromosome 3 contained 17 genes but no putative functions of interest were identified. The GK 14 days disease score QTL on chromosome 6 contained 25 genes, of which two genes, *Ca_05900* and *Ca_05885*, had potential disease resistance functions and two genes, *Ca-05898* and *Ca_05884*, had cell wall functions. The 21 days disease score QTL on chromosome 2 contained 14 genes, of which two had putative disease resistance functions.

## 4. Discussion

This study advances our understanding of the inheritance of Ascochyta resistance in chickpeas by applying quantitative trait locus methods to two hybrid chickpea families derived from a cross between a cultivated variety and two different wild crop relatives. Ascochyta resistance was evaluated under field conditions in Southern Turkey as disease scores at multiple time points and this phenotype data were analyzed with genotype data for SNPs. Three QTLs on chromosomes 2, 3, and 6 for early-stage Ascochyta resistance were identified in one family, GK, a cross with *C. echinospermum*, and one QTL on chromosome 7 for late-stage Ascochyta resistance was identified in the other family, GO, a cross with *C. reticulatum*. All QTLs had modest effect sizes of about 9 PVE with considerable dominant effects. Screening of 200k bp regions around each QTL peak identified a total of nine candidate genes with putative disease resistance or cell wall restructuring functions. 

### 4.1. Ascochyta Resistance Expression

The wild Ascochyta strain caused mild to severe symptoms across the tested lines but rarely led to plant death during the experiment, corresponding to mild to moderately aggressive type I and II pathotypes that are present in the important Middle East chickpea growing region [5]. Expression of Ascochyta resistance was similar in both families, where the cultivated parent was more susceptible than the wild parent, and family lines showed a broad range of susceptibility. Average family susceptibility was more similar to the susceptible cultivated parent but some individual lines were also as resistant or more so than the resistant parent, suggesting potential for plant breeding. Broad family resistance distributions indicate polygenic inheritance of chickpea Ascochyta resistance.

### 4.2. Ascochyta Resistance Quantitative Trait Loci

The economic impact of chickpea Ascochyta blight resistance is considerable and identification of heritable resistance is considered an important chickpea breeding objective in addition to effective agronomic practice and fungicide use [4,5,6,7]. Many mapping studies have successfully identified multiple QTLs for chickpea Ascochyta resistance across multiple linkage groups in diverse germplasm crosses [15,16,17,18].

It is difficult to compare the results of earlier mapping studies because a lack of a chickpea reference genome and diverse molecular markers used to build genetic maps makes it difficult to identify equivalent linkage groups and genomic regions across studies. The publication of the first chickpea genomes [1,2] has greatly facilitated high-throughput genotyping of genome-mapped SNPs for QTL results that can be compared between studies. Thus, more recent QTL mapping studies have identified between five and eight QTLs and clusters of QTLs mapping to all chromosomes except chromosome 5 [19,20,21]. Our QTL mapping study supports the polygenic nature of Ascochyta resistance with the identification of four QTLs on chromosomes 2, 3, 6, and 7. The positions of previously reported QTLs mostly did not overlap with those detected here, with the exception of the chromosome 2 QTL of [21] that were located within 1M bp of our peak region. Other approaches to identifying genomic regions associated with Ascochyta resistance have been enabled by the reference genome and high-throughput genotyping methods. These studies include a bulk segregant analysis study that found eleven and six QTLs for Ascochyta resistance across chromosomes 1, 2, 4, 6, and 7 in two mapping families [22] and significantly associated region for Ascochyta resistance on chromosome 4 as part of a genome-wide association study [23]. The QTL regions of this study were all distinct from these studies, highlighting the importance of examining diverse chickpea accessions to find multiple QTLs contributing to complex traits like disease resistance.

Similar to this study, others have specifically looked at alleles for Ascochyta resistance from wild accessions of *C. reticulatum* [6] and *C. echinospermum* [24,25] as part of wide crosses with cultivated chickpea. These studies of wild crop relatives found one to two QTLs for Ascochyta resistance some with large PVE of 34 to 45 [6,25]. The QTL effect sizes of the current study were relatively small from 9.04 to 9.49 PVE (Table 2), more similar to the 2.5 to 9.5 PVE observed for the *C. echinospermum* study [24]. Also similar to the *C. echinospermum* study, resistant alleles were detected for both the cultivar and wild parent. Family lines that stack multiple resistant alleles could be selected to optimize Ascochyta resistance, such as the highly resistant lines identified here in both the GO and GK families. Interestingly, all QTLs identified in this study had positive heterozygous effects from 0.44 to 0.79 (Table 2) indicative of reduced Ascochyta resistance. This might indicate negative fitness effects in hybrid chickpea progeny that should be addressed by careful selection of breeding lines. Previous studies have identified hybrid incompatibility loci in crosses between cultivars and these wild crop relatives [26]. Enhanced hypersensitive response, normally reserved for pathogen attack, is a regularly observed form of hybrid incompatibility between diverged parents [27,28].

Studies of chickpea Ascochyta resistance typically consider only a single time point. Therefore, our observation that different QTLs could be detected at different time points is important as it suggests that different genes might contribute to resistance at different stages in the infection cycle. For example, resistant varieties are typically most resistant at early developmental stages [29], while early up-regulation of defensive antioxidative enzymes was found to contribute to chickpea Ascochyta resistance [30]. Information about the timing of resistance could be combined with agronomic practice such as the timing of sowing date to minimize infective humid environmental conditions at vulnerable growing stages or adjustments to chemical applications to maximize their effectiveness [31,32].

### 4.3. Ascochyta Resistance Candidate Genes

The use of mapped SNP genotypes in this study allowed us to relate QTL peaks to defined regions of the chickpea CDC Frontier reference genome and to perform subsequent gene searches using public PulseDB genomic resources. Fourteen to 25 genes were identified in each of the 200k bp regions around the four QTL peaks on chromosomes 2, 3, 6, and 7. Gene annotations suggested at least nine genes with candidate functions in Ascochyta disease resistance. The functions of these genes were explored on their respective UniProt pages, which also provided links to related literature.

At the GO, 42 days disease score QTL on chromosome 7, gene *Ca_03156* is of particular interest because studies of its homolog in *A. thaliana* have identified it to code for a stress-response A/B barrel domain-containing protein with antifungal properties [33]. Gene *Ca_03143* is homologous with *A. thaliana RIPK* coding for a serine/threonine-protein kinase that is involved in responses to bacterial pathogens [34]. The *A. thaliana* homolog of *Ca_03139* is *TBL16, protein trichome birefringence-like 16*, coding for a potential cell wall bridging protein [35]. 

At the 14 days disease score QTL on chromosome 6, gene *Ca_05900* is homologous with *A. thaliana FBL77* coding for a F-box/LRR-repeat protein identified by UniProt similarity. Many members of this family are involved in innate plant immunity [36,37]. Gene, *Ca_05885*, has homology with *A. thaliana PLA2A* coding for a phospholipase A2-α protein, which has been implicated in the hypersensitivity response to pathogen attack [38]. Gene *Ca_05898* is homologous with *A. thaliana CAMT3* coding for a caffeoyl-CoA O-methyltransferase by UniProt similarity that is involved in cell wall reinforcement to wounding and pathogen attack. Gene *Ca_05884* is homologous to *A. thaliana GALAK* coding for galacturonokinase involved in cell wall restructuring [39].

Finally, at the 21 days disease score QTL on chromosome 2, gene *Ca_10189* is homologous to a *Glycine max* gene coding for a type III effector avirulence repeat cleavage domain-containing protein based on UniProt similarity. Finally, gene *Ca_10186* is homologous to *A. thaliana R13L1* that codes for a putative disease resistance RPP13-like NB-ARC/LRR-repeat protein 1 according to UniProt similarity.

## 5. Conclusions

In conclusion, this study reveals more about the complex polygenic nature of chickpea Ascochyta disease resistance. The genetic basis of field-based Ascochyta resistance in accessions of two wild crop relatives of chickpea, *C. reticulatum*, and *C. echinospermum*, was explored to identify four new QTLs for Ascochyta resistance. Interestingly, analysis of data representing different stages of disease progression revealed three early-acting QTLs in the GK family that would have been missed if only late-stage disease progression were examined. If the data permit, we recommend this strategy for other pathogen resistance studies as plant defense strategies might change as infection proceeds. Examination of QTL regions in the public PulseDB database, allowed the construction of candidate gene lists with putative functional annotation based on homology with model plant species. This gene list is a promising resource for future study of Ascochyta resistance in these mapping families as gene-specific markers can be developed to interrogate allelic variation in coding sequence and regulatory regions and gene expression differences can be quantified. 

## Figures and Tables

**Figure 1 genes-14-00316-f001:**
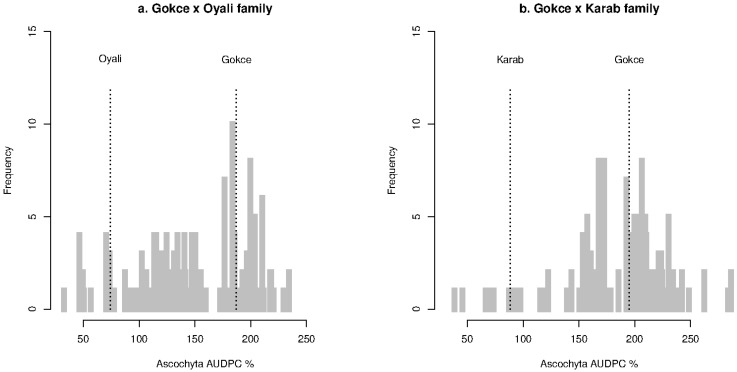
Distributions of Ascochyta area under disease curve (AUDPC) for the chickpea Gokce × Oyali (**a**) and Gokce × Karab (**b**) families. Labeled vertical dotted lines correspond to parental means for each family.

**Figure 2 genes-14-00316-f002:**
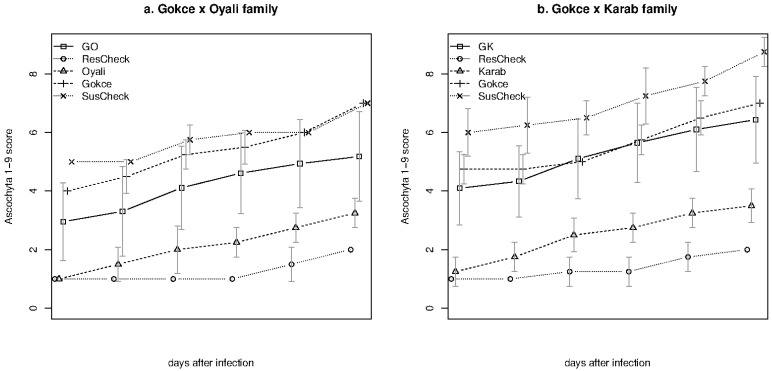
Ascochyta disease severity progression for the chickpea Gokce × Oyali (**a**) and Gokce × Karab families (**b**), parents, and check lines. Line types are Gokce × Oyali family (GO), Gokce × Karab family (GK), resistant check (ResCheck), susceptible check (SusCheck), or parental lines (Gokce, Oyali, and Karab). Plotting symbols and vertical lines show mean disease severity at each time period, respectively. Time periods for each line have been staggered slightly to better show error values.

**Figure 3 genes-14-00316-f003:**
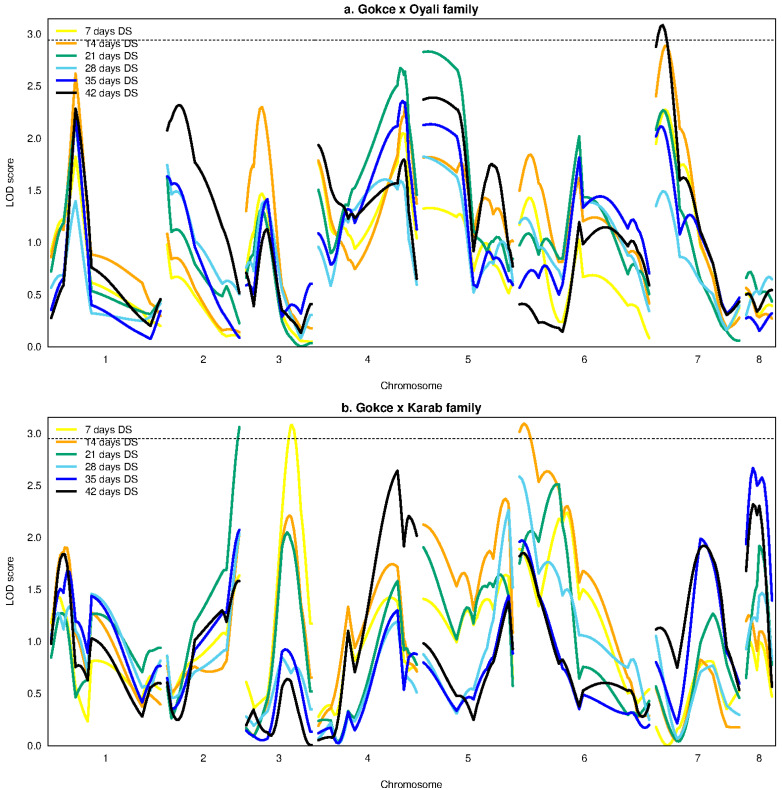
Logarithm of odds (LOD) values against genetic map position for Ascochyta disease scores at different times since inoculation for the chickpea Gokce × Oyali (panel **a**), Gokce × Karab (panel **b**) families. Each line color represents disease scores (DS) for a different time as shown in the legend. The horizontal dashed line is the LOD threshold calculated from 5000 permutations above which LOD scores indicate significant quantitative trait locus (QTL) peaks.

**Figure 4 genes-14-00316-f004:**
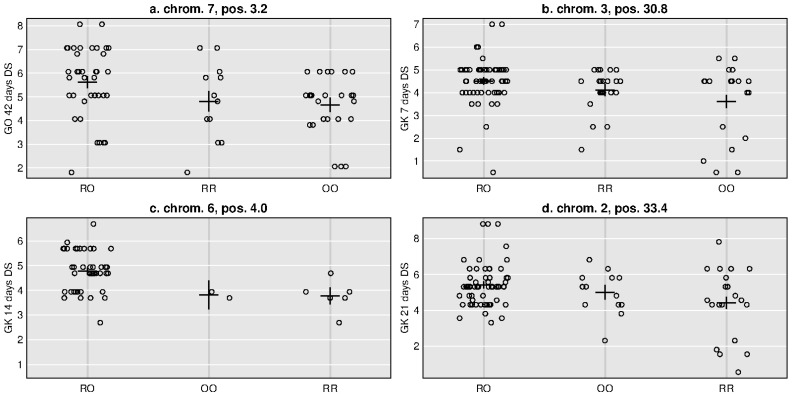
Phenotype × genotype plots for Ascochyta disease scores at various periods after inoculation representing the four quantitative trait loci (QTLs) found in two chickpeas Gokce × Oyali and Gokce × Karab families. Each figure panel is named according to QTL chromosome and Mbp position. The x-axis shows genotypes made up of cultivated reference (R) or wild other (O) alleles. Circles are per individual measures plotted with some scatter to make all values visible. Crosses are the mean and standard error per genotype.

**Table 1 genes-14-00316-t001:** Summary Ascochyta resistance measures for the chickpea Gokce × Oyali and Gokce × Karab families and check lines. Notes: Lines are the susceptible check (Sus. Check), the resistant check (Res. Check), both parental lines of each family, and the family. Sample sizes were four replicates for each check and parent, 160 lines for the GO family, and 144 lines for the GK family. Measures are disease scores (DS) at each number of days after inoculation, severity, and area under the disease progress curve (AUDPC). Measures were adjusted for block effects using augmented design. Means are followed by standard deviations in parentheses.

Line\Measure	DS 7 Days	DS 14 Days	DS 21 Days	DS 28 Days	DS 35 Days	DS 42 Days	AUDPC	Severity
	Gocke × Oyali							
Sus. Check	5 (0)	5 (0.2)	5.75 (0.68)	6 (0.24)	6 (0.24)	7 (0.13)	64.35 (2.12)	201.25 (7.57)
Res. Check	1 (0)	1 (0.2)	1 (0.2)	1 (0.24)	1.5 (0.38)	2 (0.13)	13.89 (1.87)	42 (6.64)
Gokce	4 (0)	4.5 (0.46)	5.25 (0.35)	5.5 (0.38)	6 (0.24)	7 (0.13)	59.72 (2.31)	187.25 (8.93)
Oyali	1 (0)	1.5 (0.46)	2 (0.61)	2.25 (0.31)	2.75 (0.31)	3.25 (0.38)	23.61 (3.45)	74.38 (11.98)
Family	2.96 (1.32)	3.31 (1.55)	4.11 (1.42)	4.61 (1.44)	4.94 (1.46)	5.18 (1.55)	46.49 (15.26)	147.26 (48.26)
	Gocke × Karab							
Sus. Check	6 (0.41)	6.25 (0.52)	6.5 (0.24)	7.25 (0.46)	7.75 (0.43)	8.75 (0.31)	78.7 (3.39)	245.88 (10.82)
Res. Check	1 (0.41)	1 (0.47)	1.25 (0.31)	1.25 (0.46)	1.75 (0.43)	2 (0.24)	15.28 (3.31)	47.25 (10.75)
Gokce	4.75 (0.29)	4.75 (0.13)	5 (0.38)	5.75 (0.2)	6.5 (0.24)	7 (0.24)	62.5 (1.54)	195.13 (4.71)
Oyali	1.25 (0.29)	1.75 (0.13)	2.5 (0.24)	2.75 (0.2)	3.25 (0.31)	3.5 (0.38)	27.78 (0.8)	88.38 (1.66)
Family	4.15 (1.27)	4.37 (1.21)	5.15 (1.34)	5.7 (1.39)	6.15 (1.47)	6.46 (1.47)	59.23 (13.8)	186.76 (43.76)

**Table 2 genes-14-00316-t002:** Summary of quantitative trait loci for Ascochyta resistance for the chickpea Gokce × Oyali and Gokce × Karab families. Notes. Locations are described by chromosome number, position in 1M bp units, and 1.5 logarithm of odds (LOD) confidence limits in parentheses. PVE is the percentage variance explained. SNP is marker nearest to the QTL peak. SNPs named c#.loc# were not directly genotyped but are genotype probabilities at 1M bp map intervals. Mu is the model predicted mean trait value. Genotype effect sizes on the mean phenotype are summarized as R for the reference cultivated allele and O is the other wild allele. Genotype RO is a measure of the dominance effect. The additive effect is calculated as (OO – RR)/2.

Family	Gokce × Oyali	Gokce × Karab	Gokce × Karab	Gokce × Karab
Trait	42 days DS	7 days DS	14 days DS	21 days DS
Location	7, 3.22 (0.2–19.8)	3, 30.81 (20.2–39.0)	6, 4.04 (2.0–49.6)	2, 33.36 (13.8–33.4)
LOD	3.08	3.09	3.10	3.06
PVE	9.04	9.46	9.49	9.39
SNP	c7.loc3	Ca1C19697	c6.loc4	Ca_TOG894262
Mu	4.99	4.00	4.06	4.99
RR	−0.37	0.09	−0.55	−0.71
RO	0.79	0.48	0.64	0.44
OO	−0.42	−0.57	−0.1	0.27
Add. effect	−0.03	−0.33	0.22	0.49

**Table 3 genes-14-00316-t003:** Chickpea genes with putative disease resistance-related functions found within a 200k bp region around significant Ascochyta resistance quantitative trait loci in the chickpea Gokce × Oyali and Gokce × Karab families. Notes. The reference genome and database used was CDC Frontier v1.0 in PulseDB. Identified QTL is named according to mapping family initials and DS phenotype. Name is the annotated gene name. Location is the genome browser location with Ca# identifying the chromosome and the next two numbers indicating the base pair start and end positions of the annotated gene. UniProt homology shows the top UniProt *Arabidopsis thaliana* or other species match based on SwissProt or TrEMBL searches. The species codes are: ARATH = *Arabidopsis thaliana*; and SOYBN = *Glycine max*. The top InterPro annotation is given where available.

QTL	Name	Location	UniProt Homology	InterPro Annotation
GK 21 days DS	*Ca_10189*	Ca2: 33352996 33354461	K7M9T1_SOYBN Uncharacterized protein	IPR008700: TypeIII_avirulence_cleave
GK 21 days DS	*Ca_10186*	Ca2: 33398850 33402874	R13L1_ARATH Putative disease resistance RPP13-like protein 1	IPR000767: Disease resistance protein
GK 14 days DS	*Ca_05900*	Ca6: 3993752 3995748	FBL77_ARATH F-box/LRR-repeat protein At4g29420	
GK 14 days DS	*Ca_05898*	Ca6: 3999495 4001208	CAMT3_ARATH (Probable caffeoyl-CoA O-methyltransferase At4g26220	IPR002935: SAM_O-MeTrfase
GK 14 days DS	*Ca_05885*	Ca6: 4110108 4110886	PLA2A_ARATH Phospholipase A2-α	IPR013090: PLipase_A2_AS
GK 14 days DS	*Ca_05884*	Ca6: 4112113 4121268	GALAK_ARATH Galacturonokinase	IPR000705: Galactokinase
GO 42 days DS	*Ca_03156*	Ca7: 3143105 3143773	DABB1_ARATH Stress-response A/B barrel domain-containing protein	IPR011008 Dimeric_a/b-barrel:
GO 42 days DS	*Ca_03143*	Ca7: 3236202 3238286	RIPK_ARATH Serine/threonine-protein kinase RIPK	IPR000719: Prot_kinase_dom
GO 42 days DS	*Ca_03139*	Ca7: 3293784 3297878	TBL16_ARATH Protein trichome birefringence-like 16	

## Data Availability

The data presented in this study are available in Lakmes et al. 2022 and the Appendix A.

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
