# Peer review of "Inheritance of Early and Late Ascochyta Blight Resistance in Wide Crosses of Chickpea"

_genes, 2023, doi:10.3390/genes14020316_

Round 1

Reviewer 1 Report

The current MS describes the identification of QTLs and candidate genes via analysis of two biparental populations. The overall experimental design, set-up and presentation of results is reasonable. However, following points need to be addressed for improvement of the MS.

- Provide the QTL details (early/late along with coordinates) in Table 3 and the corresponding Suppl. Table of all the genes located within QTLs.

- Details of all the QTLs identified may be presented as a Table along with number of genes identified. Why 200 kb flanking regions were also included in identification of candidate genes. Clear details about the genes identified within QTL regions and those in the flanking regions should be given.

- It is not clear why only 9 genes are selected, though they may be related to disease resistance as per GO annotation, but other genes or genes with unknown function can not be ignored for their role in disease resistance and can be better candidates.

- Overall the language of the MS is reasonable, but some improvements at few places is required.

- Section headings such as "Quantitative trait locus results " in the results section needs to be improved. Likewise in the discussion section.

Reviewer 2 Report

Dear Authors;

The work reports the inheritance of Ascochyta blight resistance in two wide crosses between the cultivar Gokce and wild chickpea accessions of C. reticulatum and C. echinospermum under field conditions in Turkey. A total of 4 QTLs were identified on the chickpea genome, and nine gene candidates involved in disease resistance and cell wall remodelling were reported. In general, the article is well written, the methodology is appropriate to achieve the objective. However, it is necessary to improve and expand the results and discussion sections regarding the inheritance of early and late Ascochyta blight resistance. It is necessary to improve the images. The article can be better with some minor comments.

Line 50-51: “Lakmes et al., 2022; in press” this format is correct?

Line 68-71: Correct subscripts in F1; F2; F2:5, etc.

Line 102-103: “Disease severity = ” must be written with formula

Line 139: Why a 200 k bp region was chosen? Explain

Line 345: 11sco???

Figure 1 and Figure 2 should improve their resolution

In the title it talks about “Inheritance of early and late Ascochyta blight resistance”.  However, within the discussion no reference is made to how these early and late identified QTLs can interact or be used. Which would be most important in a breeding program? Please include a paragraph mentioning this issue and relating it to the infection process and the genes identified.
